# Arousal-Inducing Effect of *Garcinia cambogia* Peel Extract in Pentobarbital-Induced Sleep Test and Electroencephalographic Analysis

**DOI:** 10.3390/nu13082845

**Published:** 2021-08-19

**Authors:** Duhyeon Kim, Jinsoo Kim, Seonghui Kim, Minseok Yoon, Minyoung Um, Dongmin Kim, Sangoh Kwon, Suengmok Cho

**Affiliations:** 1Department of Seafood Science and Technology, Institute of Marine Industry, Gyeongsang National University, Tongyeong 650-160, Korea; concisenews@naver.com (D.K.); jinsukim@gnu.ac.kr (J.K.); 2Research and Development Institute, S&D Co., Ltd., Cheongju 28156, Korea; so-kwon0004@hanmail.net; 3Department of Food Science and Technology, Institute of Food Science, Pukyong National University, Busan 48513, Korea; shkim.pknu@gmail.com (S.K.); hn7742@gmail.com (D.K.); 4Research Division of Food Functionality, Korea Food Research Institute, Wanju 55365, Korea; msyoon@kfri.re.kr (M.Y.); myum@kfri.re.kr (M.U.)

**Keywords:** arousal-inducing effect, caffeine, electroencephalography, *Garcinia cambogia*, psychoactive agent

## Abstract

Caffeine, a natural stimulant, is known to be effective for weight loss. On this basis, we screened the arousal-inducing effect of five dietary supplements with a weight loss effect (*Garcinia cambogia*, *Coleus forskohlii*, *Camellia sinensis* L., *Irvingia gabonensis*, and *Malus pumila* M.), of which the *G. cambogia* peel extract (GC) showed a significant arousal-inducing effect in the pentobarbital-induced sleep test in mice. This characteristic of GC was further evaluated by analysis of electroencephalogram and electromyogram in C57L/6N mice, and it was compared to that of the positive control, caffeine. Administration of GC (1500 mg/kg) significantly increased wakefulness and decreased non-rapid eye movement sleep, similar to that of caffeine (25 mg/kg), with GC and caffeine showing a significant increase in wakefulness at 2 and 6 h, respectively. Compared to that of caffeine, the shorter duration of efficacy of GC could be advantageous because of the lower possibility of sleep disturbance. Furthermore, the arousal-inducing effects of GC (1500 mg/kg) and caffeine (25 mg/kg) persisted throughout the chronic (3 weeks) administration study. This study, for the first time, revealed the arousal-inducing effect of GC. Our findings suggest that GC might be a promising natural stimulant with no side effects. In addition, it is preferential to take GC as a dietary supplement for weight loss during the daytime to avoid sleep disturbances owing to its arousal-inducing effect.

## 1. Introduction

Arousal refers to the physiological and psychological state of being awoken from sleep or sense organs stimulated to a point of perception [1,2]. It is promoted by a number of neurotransmitter systems, including acetylcholine, serotonin, norepinephrine, histamine, orexins, dopamine, and glutamate, throughout the brainstem, hypothalamus, and basal forebrain [3,4]. Insomnia, which is characterized by difficulty in initiating or maintaining sleep, has been recognized as the most common sleep disorder in recent years [5,6,7]. Excessive daytime sleepiness is a primary concern for most insomnia patients [8].

There is growing evidence that dietary factors have a relation to sleep duration and sleep quality. Consumption of healthy foods can help sleep [9]. For example, oily fish is known to be a dietary source of serotonin and vitamin D, which are involved in better sleep [10,11], while lower intake of vegetables and higher intake of soft drinks and fast food are associated with poor sleep quality [12]. Dietary components are also closely related to sleep [13]. Caffeinated coffee or tea increases sleep latency and decreases sleep duration [14].

Most people consume caffeine to compensate for a lack of sleep, increase physical performance, and boost brain activity [15,16]. Consumption of energy drinks and coffee beverages containing caffeine is increasing worldwide [17]. Caffeine (1,3,7-trimethylxanthine) is a well-known natural stimulant that induces arousal and is one of the most widely consumed psychoactive agents worldwide [18]. It has been demonstrated that caffeine induces arousal by inhibiting the adenosine 2A (A_2A_) receptor [19]. In addition, caffeine has been shown to have a weight loss effect [20,21]. It is known to induce thermogenesis and weight loss by inhibiting the phosphodiesterase-induced degradation of intracellular cyclic adenosine monophosphate [22]. In addition, caffeine aids in weight loss through the stimulation of substrate cycles, including the Cori cycle and the free fatty acid-triglyceride cycle [23,24].

In the present study, based on the fact that caffeine has both weight loss and arousal-inducing effects, we screened the arousal-inducing effects of five dietary supplements with proven weight loss effects (*Garcinia cambogia*, *Coleus forskohlii*, *Camellia sinensis* L., *Irvingia gabonensis*, and *Malus pumila* M.)*. G. cambogia* is a member of the family Clusiaceae and has (−)-hydroxycitric acid (HCA), which inhibits the synthesis of fatty acids and lipogenesis [25,26,27]. *C*. *forskohlii* previously showed a weight loss effect in ovariectomized rats [28]. Puer tea, fermented leaves of *C. sinensis* L. from Yunnan province, China, also induce an anti-obesity effect [29]. Obese subjects fed I. *gabonensis* seeds had significantly reduced total cholesterol, LDL-cholesterol, and triglycerides [30]. In a study of 71 obese male and female subjects, the group taking apple polyphenol derived from *M. pumila* M. showed a weight loss effect [31].

Among them, *G. cambogia* fruit peel extract (GC) was the most effective. The acute and chronic arousal-inducing effects and withdrawal symptoms of GC were further investigated in electroencephalogram (EEG) and electromyogram (EMG) analyses.

## 2. Materials and Methods

### 2.1. Materials

Five dietary supplements (*G. cambogia*, *C. forskohlii*, *C. sinensis* L., *I. gabonensis*, and *M. pumila* M.) were used to evaluate arousal-inducing effects. All samples were provided by Ju Yeong NS Co. Ltd. (Seoul, Korea). GC was manufactured by INDFRAG Ltd. (Bangalore, India); the product name is *Garcinia Cambogia* Extract PE-60. Following the manufacturer’s specifications, GC was extracted with demineralized water. The extraction solutions were then concentrated and dried using a spray dryer. One gram of GC contains more than 600 mg of the indicator compound, HCA. Caffeine, a reference arousal-inducing agent, was purchased from Sigma-Aldrich Inc. (St. Louis, MO, USA). The test samples were prepared by dissolving them in sterile saline containing 5% Tween 80 and were then administered to the mice via oral gavage using a sonde needle. Pentobarbital was purchased from Hanlim Pharm. Co., Ltd. (Seoul, Korea).

### 2.2. Animals

All animal experiments were conducted in accordance with the animal care and use guidelines of the Korea Food Research Institute Animal Care and Use Committee (permission number: KFRI-M-21001). The imprinting control region mice (ICR, male, 20–25 g) and C57BL/6N mice (male, 25–28 g) used in the experiments were purchased from Koatech Animal Inc. (Pyeongtaek, Korea). The animals were housed in an insulated, soundproof recording room equipped with humidity control and maintained at an ambient temperature of 23 ± 0.5 °C with an automatically controlled 12 h light/12 h dark cycle (lights on at 09:00 h) and ad libitum access to food and water. Every effort was made to minimize animal suffering and the number of animals required for the generation of reliable data.

### 2.3. Pentobarbital-Induced Sleep Test

Figure 1 shows the pentobarbital-induced sleep test, conducted according to the method described by Cho et al. [32]. All experiments were performed between 13:00 and 17:00 h, and the mice were fasted for 24 h before the experiments. The five test samples were administered orally to the ICR mice (*n* = 10) 45 min before pentobarbital administration (45 mg/kg, intraperitoneal injection (i.p.)). After the administration of pentobarbital, the mice were placed in individual cages and observed for measurements of sleep latency and duration. The observers were blinded to other treatment groups. Sleep latency was documented from the time of pentobarbital injection to the time of sleep onset, and sleep duration was defined as the difference in time between the loss and recovery of the righting reflex.

### 2.4. HPLC Analysis of Caffeine

High-performance liquid chromatography (HPLC) analysis was performed on a Hitachi L-2000 series HPLC system (Hitachi Ltd., Tokyo, Japan) consisting of an L-2130 binary pump, L-2200 autosampler, and L-2455 diode-array detector, which was set at 280 nm to monitor caffeine. A µBondapak^®^ C18 reversed-phase column, 30 cm × 3.9 mm, with a 15–20 µm particle size (Waters Ltd., Milford, MA, USA), was used at a flow rate of 1.0 mL/min. The mobile phase was composed of methanol, acetic acid, and water (20:1:79, *v/v*). The samples were prepared at a concentration of 1 mg/mL in 5% Tween 80/methanol, with the injected sample volume being 10 µL.

### 2.5. Analysis of Sleep Architecture

#### 2.5.1. Acute Administration

Both GC (500, 1000, and 1500 mg/kg) and caffeine (25 mg/kg) were administered via oral gavage to four groups of C57BL/6N mice (per group, *n* = 7–8) at 09:00 h on the day of experiment.

#### 2.5.2. Chronic Administration

The experimental procedure for chronic administration is shown in Figure 2a. GC (1500 mg/kg) and caffeine (25 mg/kg) were administered orally (p.o.) to the C57BL/6N mice (each group, *n* = 7–8) daily at 09:00 h. The administration lasted for 23 consecutive days. The EEG and EMG were recorded at baseline (BL), on the 1st, 7th, 14th, and 21st days, and in the withdrawal period.

#### 2.5.3. Polysomnographic Recordings and Vigilance State

The polysomnographic recordings were conducted using the method of Yang et al. [5]. To record polysomnographic signals, a head mount (#8201; Pinnacle Technology, Inc., Lawrence, KS, USA) equipped with EEG and EMG electrodes was chronically implanted into C57BL/6N mice (Figure 2b). The mice were anesthetized by pentobarbital injection (50 mg/kg, i.p.). Anesthetized mice were shaved prior to surgery after their heads and necks were cleaned with 70% alcohol. The front edge of the head mount was placed on the skull, positioned 3.0 mm anterior to the bregma. Then, four electrode screws were inserted in holes perforated into the skull. The double wire electrodes were inserted on both sides of small pockets made into the nuchal muscles. To fix the head mount on the skull, dental cement was used. After surgery, the mice were transferred to separate cages to recover at least for 1 week. Subsequently, they were habituated to the recording conditions during the 3–4 days before the experiments. The EEG and EMG recordings were performed using a slip ring designed not to restrict the movement of the mouse. The EEG and EMG signals were recorded using the PAL-8200 data acquisition system (Pinnacle Technology Inc., Lawrence, KS, USA). The EEG and EMG signals were amplified 100-fold and low-pass filtered at 25 Hz for EEG and 100 Hz for EMG. All signals were stored at a sampling rate of 200 Hz. The sleep–wake states, including baseline data acquisition and experiments, were monitored for a period of 48 h. Baseline recordings were taken for each mouse over the course of 24 h (beginning at 09:00 h) and served as a control for the same mouse. The vigilance states were automatically classified by a 10 s epoch as wakefulness, non-rapid eye movement sleep (NREMS), or rapid eye movement sleep (REMS) by SleepSign version 3.0 (Kissei Comtec, Nagano, Japan). Sleep–wake stages were checked visually and corrected if needed. During NREMS, the delta activity in the 0.5–4 Hz range was first summed and normalized as a percentage of the corresponding average delta power of NREMS. Bouts of wakefulness, NREMS, and REMS were defined as periods of one or more consecutive 10 s epochs (Figure 3).

### 2.6. Statistical Analysis

All data were expressed as the mean ± standard error of mean. Statistical analysis was performed using Prism 8.0 (GraphPad Software Inc., San Diego, CA, USA). For performing multiple comparisons, the data were analyzed using one-way analysis of variance followed by Dunnett’s test. The two groups of data were compared using an unpaired Student’s *t*-test. *p*-values of less than 0.05 were considered significant for all statistical tests.

## 3. Results

### 3.1. Screening of the Arousal-Inducing Effect of Dietary Supplements with Weight Loss Effect

To evaluate the arousal-inducing effects of dietary supplements with weight loss effects, we first conducted a pentobarbital-induced sleep test in ICR mice (Figure 3a). Caffeine, a well-known stimulant, was used as a positive control. The sleep latency and duration in the control group were 212.3 ± 3.9 s and 61.5 ± 2.9 min, respectively. As expected, the positive control caffeine (50 mg/kg, p.o.) produced a significant arousal-inducing effect by resulting in an increase in sleep latency (237.7 ± 5.5 s, *p* < 0.01) and decrease in sleep duration (41.9 ± 3.1 min, *p* < 0.001) as compared to those in the control group. Among the dietary supplements, only GC and *C. sinensis* L. (at 1000 mg/kg each) exhibited arousal-inducing effects. In particular, GC showed significant arousal-inducing effects on both sleep latency and sleep duration.

In the next stage, dose-dependent effects (500, 1000, and 1500 mg/kg) of GC were tested (Figure 3b). GC resulted in a significant increase in sleep latency at all dosages; however, a significant decrease in sleep duration was observed at 1000 and 1500 mg/kg. The results demonstrated that GC has an arousal-inducing effect similar to that of caffeine.

### 3.2. Effects of GC on Sleep–Wake Profiles and Time-Course Changes in C57BL/6N Mice

To better understand the arousal-inducing effect of GC, we analyzed the total amount of time spent in wakefulness and non-rapid eye movement sleep (NREMS) during the first 2 h following the administration of caffeine and GC. Caffeine (25 mg/kg) increased wakefulness by 1.9-fold (*p* < 0.001) compared to that of the vehicle (Figure 4a). Meanwhile, GC (500, 1000, and 1500 mg/kg) showed a dose-dependent increase in arousal induction. In addition, the effect of GC at 1500 mg/kg was comparable to that of caffeine at 25 mg/kg, with no significant difference between them.

The time-course changes in wakefulness and NREMS for 24 h after the administration of caffeine and GC (at 09:00 h) are shown in Figure 4b,c, respectively. The significant arousal-inducing effect of caffeine (25 mg/kg) lasted for 6 h after administration (Figure 4b), whereas GC (1500 mg/kg) produced a significant increase in wakefulness that lasted for approximately 2 h (Figure 4c). The decrease in NREMS was accompanied by an increase in wakefulness. There was no further disruption in the sleep architecture during the subsequent period.

### 3.3. Effects of GC on the Characteristics of Sleep–Wakefulness Episodes and Power Density

To determine the arousal-inducing effects of GC, we evaluated the mean duration and total number of NREMS, REMS, and wakefulness episodes. It was seen that GC (1500 mg/kg) and caffeine (25 mg/kg) significantly increased the mean duration of wakefulness episodes by 3.0-fold (*p* < 0.01) and 3.7-fold (*p* < 0.01), respectively (Figure 5a). In addition, they produced a significant decrease in the number of wakefulness bouts (GC: 1.7-fold, *p* < 0.01; caffeine: 2.5-fold, *p* < 0.01) in the 2 h after administration (Figure 5b).

To evaluate sleep quality, delta activity (frequency range of 0.5–4 Hz) was analyzed from the EEG power density in mice during NREMS. When compared to that of the vehicle, both caffeine and GC resulted in no differences in EEG power density, including delta activity, in NREMS (Figure 5c).

### 3.4. Arousal-Inducing Effect of GC in the Chronic (3 Weeks) Administration Study

To identify tolerance and withdrawal symptoms associated with the arousal-inducing effect of GC, a chronic (3 weeks) administration test was performed. It was seen that wakefulness and NREMS in both caffeine and GC groups recovered to the same levels as that of the BL group on the withdrawal day (WD). There was no significant difference between the results of the WD and BL for both the groups (Figure 6). From the results, it was confirmed that chronic (3 weeks) administration of GC and caffeine does not induce tolerance or withdrawal symptoms in mice.

### 3.5. Caffeine Content of GC

To demonstrate whether caffeine contributed to the arousal-inducing effect of GC, the caffeine content of GC was measured by HPLC analysis. The standard caffeine was detected at 18 min, which was not detected in the GC sample (Figure 7). This result suggests that GC induces arousal through active compounds other than caffeine.

## 4. Discussion

In the present study, we confirmed for the first time the arousal-inducing effect of GC by screening dietary supplements with a weight loss effect. *G. cambogia* is used in traditional medicines in many Asian countries to treat various diseases, including intestinal parasites, constipation, bowel complaints, rheumatism, edema, and delayed menstruation [33]. In addition, it possesses anticancer, antidiabetic, antioxidant, anti-inflammatory, antiulcer, antimicrobial, antineoplastic, and gastroprotective activities [34,35,36,37,38,39]. In particular, the extract obtained from *G. cambogia* fruit peel has been commercialized as a dietary supplement for weight loss [27,40].

The Korean Ministry of Food and Drug Safety has approved the use of GC as a dietary supplement, and its recommended dosage is 750–2800 mg/day [41]. In addition, the effects of GC (between 1000 and 3000 mg/day) were evaluated in weight loss clinical trials [42]. In this study, caffeine (25 mg/kg) and GC (1500 mg/kg) showed similar arousal-inducing effects. Since there is an average of 50 mg of caffeine in a 100 mL cup of coffee, the stimulative effects of GC can be expected when it is administered at doses of over 3000 mg in humans [43,44].

Because rodents are nocturnal animals, it is not appropriate to evaluate the arousal-inducing effect of these animals during the evening [45]. Therefore, GC and caffeine were administered to the mice during the daytime (at 09:00 h). The arousal-inducing effect of caffeine (25 mg/kg) persisted for up to 6 h after administration (Figure 4b). The long duration of caffeine efficacy was observed in the study by Huang et al. [19]. A previous study also reported that caffeine can trigger a decrease in the ability to develop or maintain deeper stages of NREMS [46]. In our study, GC (1500 mg/kg) maintained the arousal-inducing effect for up to 2 h after administration (Figure 4c). The shorter duration of efficacy of GC could be advantageous for night sleep [47,48]. Delta activity is an indicator of the intensity or depth of NREMS [49,50]. In a previous study by Kwon et al. [51], caffeine was reported to maintain sleep quality with no change in delta activity. In this study, both caffeine (25 mg/kg) and GC (1500 mg/kg) showed no changes in delta activity during NREMS (Figure 5c).

To date, research on the arousal-inducing effects of various plant extracts has been limited to acute administration [52]. Therefore, in the present study, we also confirmed that the arousal-inducing effect of GC is maintained during the chronic administration period (3 weeks; Figure 6). Oral administration of GC (1500 mg/kg) significantly increased wakefulness and decreased NREMS on the first day, and this activity continued throughout the administration period. In addition, GC did not produce tolerance during the chronic administration period, nor did it cause any adverse effects during withdrawal. These results imply that GC might be potentially used as a natural stimulant without any adverse effects related to the sleep–wakefulness cycle.

To confirm whether the arousal-inducing effect of GC was due to caffeine, the caffeine content of GC was measured. The results showed that GC did not contain caffeine, which was consistent with the results of a previous study [53]. Furthermore, HCA has been recognized as the major active constituent and indicator compound of GC [54]. The content of HCA in the rind of *G. cambogia* ranges between 20% and 60% [40,55]. Though HCA has been demonstrated to be effective in improving various conditions such as inflammation, oxidative stress, and insulin resistance [56,57], the arousal-inducing effects of HCA have not yet been investigated.

## 5. Conclusions

To the best of our knowledge, this is the first study to demonstrate the arousal-inducing effect of GC, which might be a promising arousal-inducing agent without caffeine. When considering the arousal-inducing effect of GC, it would be desirable to take GC during the daytime, rather than in the evening, for weight loss. Further studies are needed to investigate the mechanism through which GC induces arousal and the arousal-inducing effect of HCA, the major active compound of GC.

## Figures and Tables

**Figure 1 nutrients-13-02845-f001:**
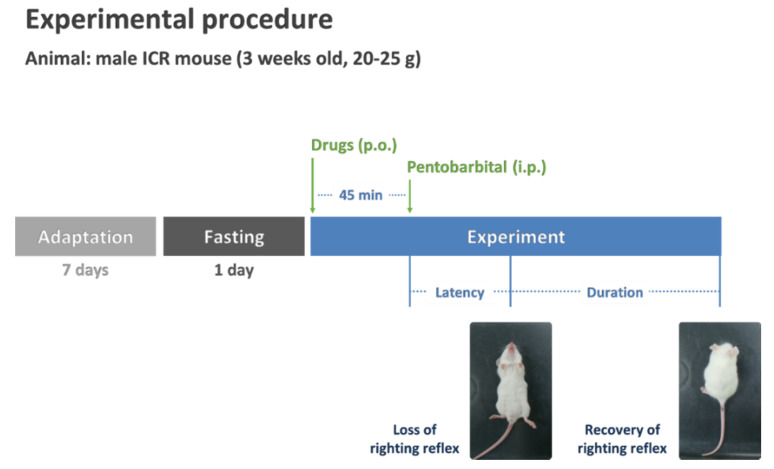
Protocol of pentobarbital-induced sleep test in ICR mice. ICR, imprinting control region; i.p., intraperitoneal injection; p.o., per os injection.

**Figure 2 nutrients-13-02845-f002:**
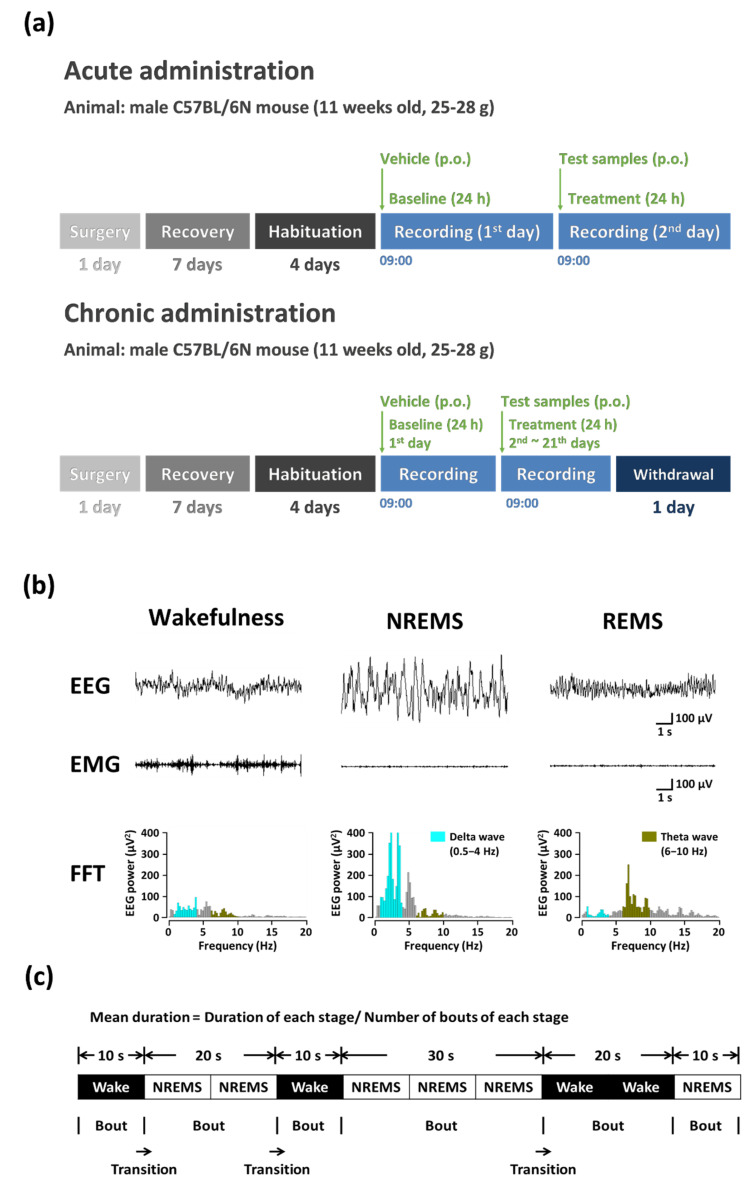
(**a**) Experimental procedure of acute and chronic administration for polysomnographic recordings. (**b**) Typical EEG, EMG, and FFT spectra in C57BL/6N mice. (**c**) Definition of sleep–wake episodes. p.o., per os injection. NREMS, non-REMS; REMS, rapid eye movement sleep; EEG, electroencephalogram; EMG, electromyogram; FFT, fast Fourier transform.

**Figure 3 nutrients-13-02845-f003:**
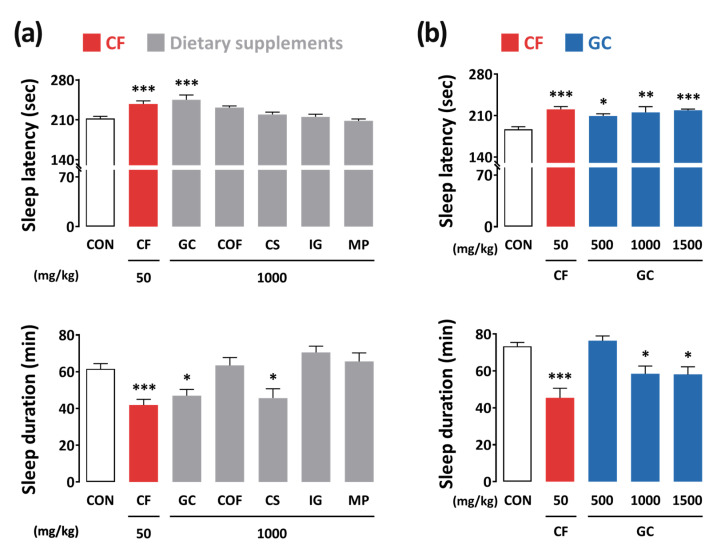
(**a**) Effects of dietary supplements on sleep latency and sleep duration in ICR mice treated with a hypnotic dose (45 mg/kg, i.p.) of pentobarbital. (**b**) Effects of GC and caffeine on sleep latency and sleep duration in ICR mice. The mice received pentobarbital injection 45 min after the oral administration (p.o.) of samples. Each column represents the mean ± SEM (*n* = 10). * *p* < 0.05, ** *p* < 0.01, *** *p* < 0.001, significant difference as compared to the control group (Dunnett’s test). CON, control; CF, caffeine; GC, *G. cambogia* peel extract; COF, *C. forskohlii* extract; CS, *C. sinensis* L. extract; IG, *I. gabonensis* extract; MP, *M. pumila* Mill. extract; SEM, standard error of mean.

**Figure 4 nutrients-13-02845-f004:**
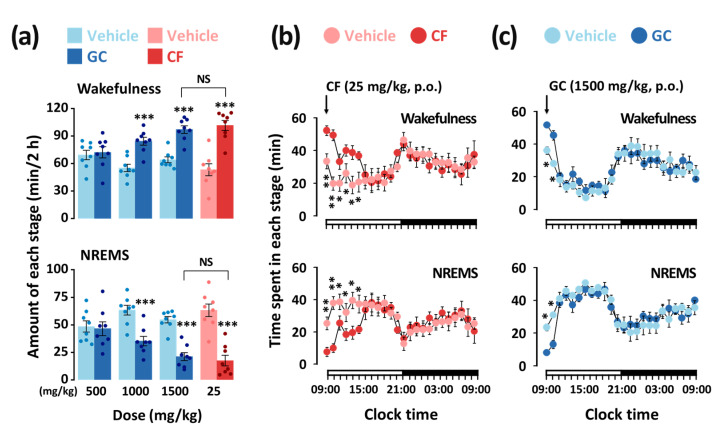
(**a**) Effects of GC and caffeine on wakefulness and NREMS during the 2 h period after administration. Effects of caffeine (**b**) and GC (**c**) on time-course changes in wakefulness and NREMS over a period of 24 h. Light color bars indicate the baseline day (vehicle). Light and deep color circles indicate the baseline day (vehicle) and experimental day (caffeine or GC), respectively. Each circle represents the hourly mean ± SEM (*n* = 7–8) of wakefulness and NREMS. Each value represents the mean ± SEM of each group. (*n* = 7–8). * *p* < 0.05, ** *p* < 0.01, *** *p* < 0.001, significantly different from the vehicle (unpaired Student’s *t*-test). CF, caffeine; GC, *G. cambogia* peel extract; NREMS, non-rapid eye movement sleep; SEM, standard error of mean; NS, no significant difference.

**Figure 5 nutrients-13-02845-f005:**
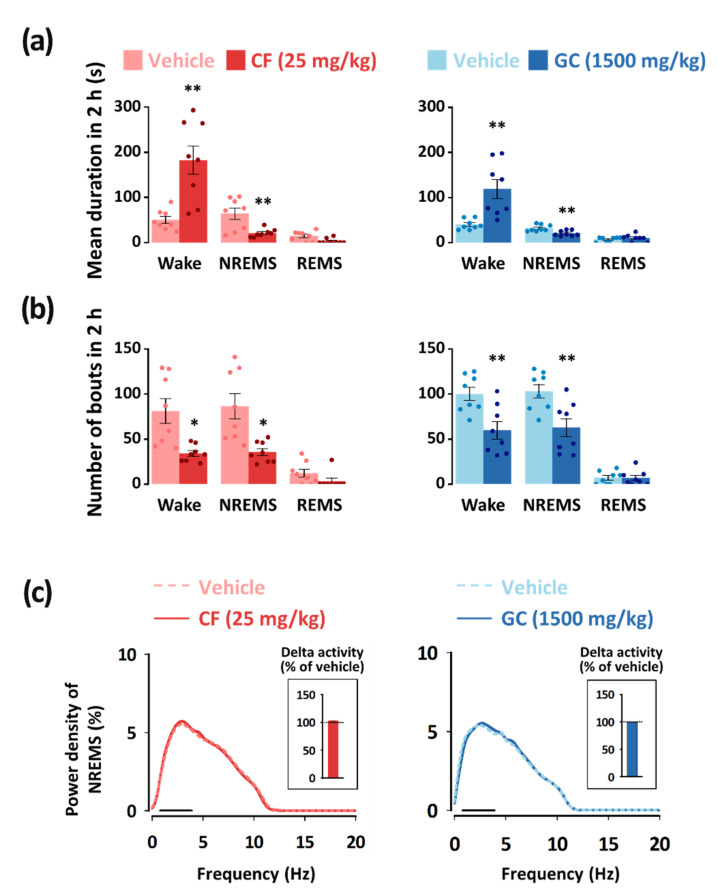
Characteristics of sleep–wake bouts in C57BL/6N mice during the 2 h period after administration of caffeine (25 mg/kg) and GC (1500 mg/kg). (**a**) Changes in the mean duration of wakefulness, NREMS, and REMS bouts. (**b**) Changes in the total number of wakefulness, NREMS, and REMS bouts. (**c**) EEG power density curves during NREMS. Light color bars indicate the baseline day (vehicle). Delta activity, an indicator of sleep intensity, is shown in the inset histogram. The solid bar (—) represents the range of the delta wave (0.5–4 Hz). Each value represents the mean ± SEM of each group. (*n* = 7–8). * *p* < 0.05, ** *p* < 0.01, significantly different from vehicle (unpaired Student’s *t*-test). CF, caffeine; GC, *G. cambogia* peel extract; NREMS, non-REMS; REMS, rapid eye movement sleep; Wake, wakefulness; SEM, standard error of mean.

**Figure 6 nutrients-13-02845-f006:**
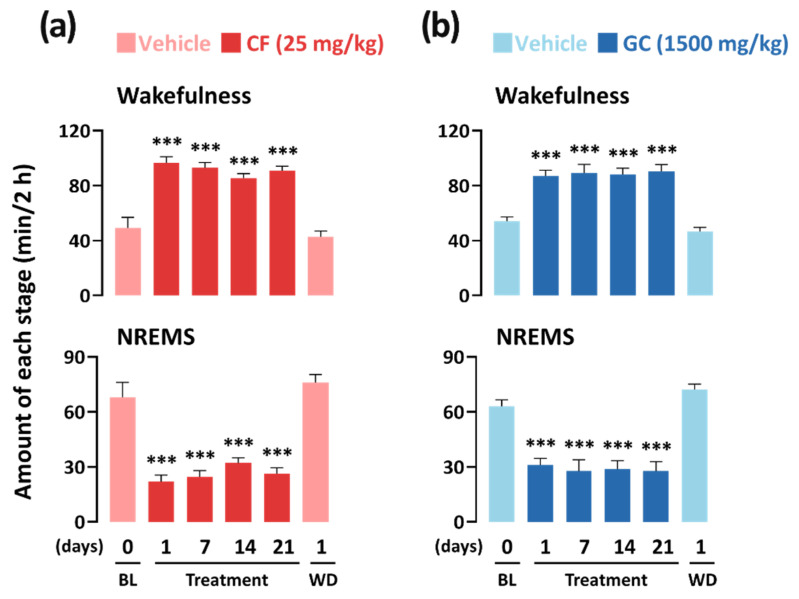
Chronic (3 weeks) administration effects of caffeine (**a**) and GC (**b**) on wakefulness and NREMS in C57BL/6N mice over a period of 2 h after administration. Light color bars indicate the baseline (BL, vehicle) or withdrawal (WD) day. Each value represents the mean ± SEM of each group. (*n* = 7–8). *** *p* < 0.001, significantly different from vehicle (unpaired Student’s *t*-test). CF, caffeine; GC, *G*. *cambogia* peel extract; NREMS, non-REMS; Wake, wakefulness; SEM, standard error of mean.

**Figure 7 nutrients-13-02845-f007:**
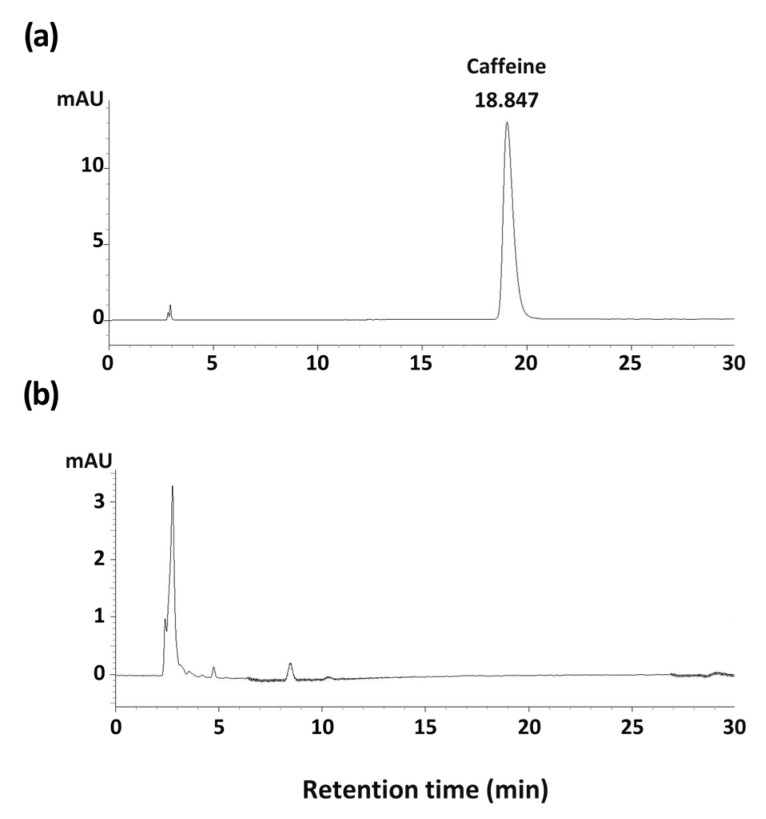
HPLC chromatogram of a caffeine reference standard (**a**) and GC (**b**). GC, *G*. *cambogia* peel extract.

## Data Availability

Not applicable.

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
