# Peer review of "Arousal-Inducing Effect of Garcinia cambogia Peel Extract in Pentobarbital-Induced Sleep Test and Electroencephalographic Analysis"

_nutrients, 2021, doi:10.3390/nu13082845_

Round 1

Reviewer 1 Report

Kim et al.’s manuscript entitled “Arousal-Inducing Effect of Garcinia cambogia Peel Extract in Pentobarbital-Induced Sleep Test and Electroencephalographic Analysis” is interesting. The authors showed the effects of G. cambogia, compared to C. forskholii, C. sinensis L., I. gabonensis, M. pumila, and caffeine. To increase the impact and quality of this paper, I have the following concerns.

Major:

  1. Please include a brief intro of each dietary supplement used in the study such as G. cambogia, C. forskohlii, C. sinensis L., I. gabonensis, and M. pumila in section 1. introduction.
  2. The conclusion section needs to be rewritten.  
  3. Please add citations in the methods section.  

Minor:

  1. ‘Stimulant’ is not a keyword, remove it or specify. Rearrange the keywords in alphabetical order.
  2. Please recheck the ‘figure 7’ caption.
  3. Section 2.1. revise plant names as G. cambogia, C.forskholii, C. sinensis L., I. gabonensis, and M. pumila.
  4. Section2.1. please specify the extraction manufacturers’ protocol, for instance, protocol version. 

Author Response

Reviewer 1

General comments: Kim et al.’s manuscript entitled “Arousal-Inducing Effect of Garcinia cambogia Peel Extract in Pentobarbital-Induced Sleep Test and Electroencephalographic Analysis” is interesting. The authors showed the effects of G. cambogia, compared to C. forskohlii, C. sinensis L., I. gabonensis, M. pumila M., and caffeine. To increase the impact and quality of this paper, I have the following concerns.

Answer: We thank the reviewer for the kind review of our study. We have revised the manuscript according to the comments. We are sure that the comments have greatly improved the quality of our paper.

[Major comments]

 Comment 1: Please include a brief intro of each dietary supplement used in the study such as G. cambogia, C. forskohlii, C. sinensis L., I. gabonensis, and M. pumila M. in section 1. introduction.

Answer: We appreciate the expert opinion. In response to the reviewer’s comment, we added a brief intro of five dietary supplements (G. cambogia, C. forskohlii, C. sinensis L., I. gabonensis, and M. pumila M.).

[Introduction section: P2, L55-65]

 In the present study, based on the fact that caffeine has both weight loss and arousal-inducing effects, we screened the arousal-inducing effects of five dietary supplements (Garcinia cambogia, Coleus forskohlii, Camellia sinensis L., Irvingia gabonensis, and Malus pumila M.) with proven weight loss effects. G. cambogia is a member of the family Clusiaceae and has (−)-hydroxycitric acid (HCA), which inhibits the synthesis of fatty acids and lipogenesis [19-21]. C. forskohlii previously showed a weight loss effect in ovariectomized rats [22]. Pu'er tea, fermented leaves of C. sinensis L. from Yunnan province, China, also induce an anti-obesity effect [23]. Obese subjects fed I. gabonensis seeds had significantly reduced total cholesterol, LDL-cholesterol, and triglycerides [24]. In a study of 71 obese male and female subjects, the group taking apple polyphenol derived from M. pumila M. showed a weight loss effect [25].

Comment 2: The conclusion section needs to be rewritten.

Answer: In response to the reviewer’s comment, we rewrote the conclusion section. We have revised and re-edited to ensure correct English.

[Conclusion section: P12-13, L306-311]

 To the best of our knowledge, this is the first study to demonstrate the arousal-inducing effect of GC. GC might be a promising arousal-inducing agent without caffeine. When considering the arousal-inducing effect of GC, it would be desirable to take GC during the daytime, rather than in the evening, for weight loss. Further studies are needed to investigate the mechanism through which GC induces arousal and the arousal-inducing effect of HCA, the major active compound of GC.

Comment 3: Please add citations in the methods section.

Answer: In response to the reviewer’s comment, we added citation of the polysomnographic recordings and vigilance state in the methods section.

[Materials and methods section: P4, L129-130]

 The polysomnographic recordings were conducted using the method of Yang et al. [5].

[Reference]

  1. Yang, H.; Yoon, M.; Um, M.Y.; Lee, J.; Jung, J.; Lee, C.; Kim, Y.-T.; Kwon, S.; Kim, B.; Cho, S. Sleep-promoting effects and possible mechanisms of action associated with a standardized rice bran supplement. Nutrients 2017, 9, 512.

[Minor comments]

 Comment 1: ‘Stimulant’ is not a keyword, remove it or specify. Rearrange the keywords in alphabetical order.

Answer: In response to the reviewer’s comment, we removed ‘stimulant’ and replaced with ‘psychoactive agent’. In addition, we rearranged the keywords in alphabetical order.

[Keywords section: P1, L32-33]

 Keywords: Arousal-inducing effect; caffeine; electroencephalography; Garcinia cambogia; psychoactive agent.

Comment 2: Please recheck the ‘figure 7’ caption.

Answer: In response to the reviewer’s comment, we clarified the expression because we performed HPLC analysis using caffeine reference standard to investigate the caffeine content of GC.

 [Results section: P10, L257]

Figure 7. HPLC chromatogram of a caffeine reference standard (a) and GC (b). GC, G. cambogia peel extract.

Comment 3: Section 2.1. revise plant names as G. cambogia, C.forskohlii, C. sinensis L., I. gabonensis, and M. pumila M.

Answer: As the reviewer’s comments, we revised plant names.

[Materials and methods section: P2, L71-72]

 Five dietary supplements (G. cambogia, C. forskohlii, C. sinensis L., I. gabonensis, and M. pumila M.) were used to evaluate the arousal-inducing effects.

Comment 4: Section2.1. please specify the extraction manufacturers’ protocol, for instance, protocol version.

Answer: As the reviewer’s comments, we specified the extraction manufacturer’s protocol.

[Materials and methods section: P2, L73-77]

 GC was manufactured by INDFRAG Ltd. (Bangalore, India); the product name is Garcinia Cambogia Extract PE-60. Following the manufacturer’s specifications, GC was extracted with demineralized water. The extraction solutions were then concentrated and dried using a spray dryer. One gram of GC contains more than 600 mg of the indicator compound, HCA

Reviewer 2 Report

Introduction

Introduction must be improved as it gives the feeling that the study will focus on caffeine. I feel focusing two paragraphs on caffeine effect is not suitable as it was used only as positive control in this study.

I suggest authors to better elaborate the part regarding dietary factors, which were shown to affect both sleep latency and sleep duration (explored also in this study) PMID: 33549913.

Methods

Authors should provide HPLC analyses or report the content of explored supplements, or if the information is not available provide an adequate statement in the limitations section. Further information on five supplements should be also provided.

Discussion

Discussion should be entirely rephrased.

First paragraph in the discussion reports conflicting statements. First authors sate they confirmed the arousal-inducing effect of GC, then by the end of the paragraph they report it was the first study to explore such an effect - please rephrase.

Second paragraph should be deleted as this is discussion section.

Third paragraph does not make any sense, as caffeine was used as positive control, not as studied compound.

Other studies exploring the arousal- inducing effect of GC or other plant extracts should be better discussed, together with possible mechanisms underlying such effect.

Author Response

Reviewer 2

 Introduction

 Comment 1: Introduction must be improved as it gives the feeling that the study will focus on caffeine. I feel focusing two paragraphs on caffeine effect is not suitable as it was used only as positive control in this study. I suggest authors to better elaborate the part regarding dietary factors, which were shown to affect both sleep latency and sleep duration (explored also in this study) PMID: 33549913.

Answer: We appreciate the comment and fully agree. According to the reviewer’s comment, unnecessary information on caffeine has been deleted, because caffeine was just used as a positive control.

[Introduction section: P2, L47-54]

 Caffeine (1,3,7-trimethylxanthine) is a well-known natural stimulant that induces arousal, and is one of the most widely consumed psychoactive agents worldwide [12]. It has been demonstrated that caffeine induces arousal by inhibiting the adenosine 2A (A2A) receptor [13]. In addition, caffeine has been shown to have a weight loss effect, in addition to the arousal-inducing effect [14,15]. It is known to induce thermogenesis and weight loss by inhibiting the phosphodiesterase-induced degradation of intracellular cyclic adenosine monophosphate [16]. In addition, caffeine aids in weight loss through the stimulation of substrate cycles, including the Cori cycle and the free fatty acid-triglyceride cycle [17,18].

Materials and Methods

Comment 2: Authors should provide HPLC analyses or report the content of explored supplements, or if the information is not available provide an adequate statement in the limitations section. Further information on five supplements should be also provided.

 Answer: We fully agree for your comment. However, we have focused on the arousal-inducing effects of GC because Garcinia cambogia was the most effective among five dietary supplements. Hence, the caffeine content of GC was additionally analyzed. We would appreciate it if you understand.

Discussion

 Comment 3: Discussion should be entirely rephrased.
First paragraph in the discussion reports conflicting statements. First authors sate they confirmed the arousal-inducing effect of GC, then by the end of the paragraph they report it was the first study to explore such an effect - please rephrase.

Answer: In response to reviewer’s comment, we rephrased first paragraph.

[Discussion section: P11, L260-261]

 In the present study, we confirmed for the first time the arousal-inducing effect of GC by screening dietary supplements with a weight loss effect.

Comment 4: Second paragraph should be deleted as this is discussion section.

Answer: In response to reviewer’s comment, we deleted second paragraph. Instead, a reason for evaluating the arousal-inducing effect in rodents during the evening was added to the third paragraph.

[Discussion section: P11, L275-278]

 Because rodents are nocturnal animals, it is not appropriate to evaluate the arousal-inducing effect of these animals during the evening [39]. Therefore, GC and caffeine were administered to the mice during the daytime (09:00 h). The arousal-inducing effect of caffeine (25 mg/kg) was persisted for up to 6 h after administration (Figure 4b).

Comment 5: Third paragraph does not make any sense, as caffeine was used as positive control, not as studied compound.

Answer: In response to reviewer’s comment, we deleted third paragraph.

Comment 6: Other studies exploring the arousal-inducing effect of GC or other plant extracts should be better discussed, together with possible mechanisms underlying such effect.

Answer: We appreciate the comment and fully agree. We added that further studies on the mechanism of arousal action of GC should be needed in the conclusion section.

[Conclusion section: P11-12, L309-311]

 Further studies are needed to investigate the mechanism of through which GC induces and the arousal-inducing effect of HCA, the major active compound of GC.

Reviewer 3 Report

Dear Authors, a properly written review of the literature, methodology and research methods described in an understandable and clear manner, clearly presented results, well-chosen research statistics, abundant discussion of the results. A clear conclusion. Correct literature on the subject. 

Author Response

Reviewer 3

General Comment : Dear Authors, a properly written review of the literature, methodology and research methods described in an understandable and clear manner, clearly presented results, well-chosen research statistics, abundant discussion of the results. A clear conclusion. Correct literature on the subject.

Answer: We thank you so much for your comment.

Reviewer 4 Report

The manuscript entitled “Arousal-Inducing Effect of Garcinia cambogia Peel Extract in Pentobarbital-Induced Sleep Test and Electroencephalographic Analysis” (ID nutrients-1330607) by Kim et al. is a well written and interesting paper presenting results of evaluation of the arousal-inducing effect of five dietary supplements including Garcinia cambogia. The arousal-inducing effect of GC a major novelty of presented paper. It has been reported here for the first time. This article will be of interest to many in the field of utilization of natural products in the supplementation.

However, there are some points that could be improved:

  • Authors suggest that for observed effects is responsible GC. Did the authors perform some preliminary studies to confirm the presence of GC in the blood plasma of animals with the use of reported HPLC method or LC-MS/MS method cited in the reference 53 (Viana et al., 2018)? If there are such results it would be worthy to implement appropriate information in the discussion part of the manuscript.
  • Some details in the methods section are missing and should be provided, namely page 10: the chromatographic peak of GC on Figure 7b should be indicated and retention time should be given?

Author Response

Reviewer 4

General Comment : The manuscript entitled “Arousal-Inducing Effect of Garcinia cambogia Peel Extract in Pentobarbital-Induced Sleep Test and Electroencephalographic Analysis” (ID nutrients-1330607) by Kim et al. is a well written and interesting paper presenting results of evaluation of the arousal-inducing effect of five dietary supplements including Garcinia cambogia. The arousal-inducing effect of GC a major novelty of presented paper. It has been reported here for the first time. This article will be of interest to many in the field of utilization of natural products in the supplementation.

Answer: We thank the reviewer for the kind review of our study. We have revised the manuscript according to the comments.

Comment : However, there are some points that could be improved:
Authors suggest that for observed effects is responsible GC. Did the authors perform some preliminary studies to confirm the presence of GC in the blood plasma of animals with the use of reported HPLC method or LC-MS/MS method cited in the reference 53 (Viana et al., 2018)? If there are such results it would be worthy to implement appropriate information in the discussion part of the manuscript.

 Answer: We appreciate the comment. We did not confirm the presence of GC in blood plasma of animals, but only confirmed the arousal-inducing effects of GC in animal experiments.

Comment : Some details in the methods section are missing and should be provided, namely page 10: the chromatographic peak of GC on Figure 7b should be indicated and retention time should be given?

 Answer: We fully agree with the reviewer and appreciate the expert opinion. We added the retention time “18.847” in Figure 7a. The peak shown in Figure 7b is the solvent peak used for HPLC analysis.

Round 2

Reviewer 2 Report

Authors addressed most of the comments. Nonetheless, few sentences introducing the effect of dietary factors toward sleep duration and quality should be added (please see also previous comment).

Author Response

Introduction

 Comment 1: Authors addressed most of the comments. Nonetheless, few sentences introducing the effect of dietary factors toward sleep duration and quality should be added (please see also previous comment).

Answer: We appreciate the comment and fully agree. According to the reviewer’s comment, we added few sentences introducing the effect of dietary factors toward sleep duration and quality.

[Introduction section: P1-2, L43-48]

 There is growing evidences that dietary factors have a relation to sleep duration and sleep quality. Consumption of healthy foods can help sleep [9]. For example, oily fish is known to a dietary source of serotonin and vitamin D, which are involved in better sleep [10,11]. While, lower intake of vegetables and higher intake of soft drinks and fast food is associated with poor sleep quality [12]. Dietary components are also closely related to sleep [13]. Caffeinated coffee or tea increase sleep latency and decrease sleep duration [14].
